# VespAI: a deep learning-based system for the detection of invasive hornets

Thomas A. O'Shea-Wheller [1] ✉, Andrew Corbett[2], Juliet L. Osborne [1], Mario Recker[3,4] &
Peter J. Kennedy [1]

The invasive hornet *Vespa velutina nigrithorax* is a rapidly proliferating threat to pollinators in Europe and East Asia. To effectively limit its spread, colonies must be detected and destroyed early in the invasion curve, however the current reliance upon visual alerts by the public yields low accuracy. Advances in deep learning offer a potential solution to this, but the application of such technology remains challenging. Here we present VespAI, an automated system for the rapid detection of *V. velutina*. We leverage a hardware-assisted AI approach, combining a standardised monitoring station with deep YOLOv5s architecture and a ResNet backbone, trained on a bespoke end-to-end pipeline. This enables the system to detect hornets in real-time—achieving a mean precision-recall score of ≥0.99—and send associated image alerts via a compact remote processor. We demonstrate the successful operation of a prototype system in the field, and confirm its suitability for large-scale deployment in future use cases. As such, VespAI has the potential to transform the way that invasive hornets are managed, providing a robust early warning system to prevent ingressions into new regions.

The detection of invasive species at the earliest possible juncture is crucial in mitigating their impacts, and often represents the only feasible opportunity to prevent population establishment[1,2]. This challenge has traditionally been approached via manual surveying, trapping, and predictive modelling[3–5], however such tools face substantial limitations when applied to small and mobile social insects, which represent ~40% of the most successful invasive invertebrates globally[6]. Specifically, the difficulty of detecting repeated but rare queen dispersal and founding events[7,8], combined with the exponential growth of colony populations from initial bridgeheads[7–9], severely impedes the efficacy and cost-effectiveness of manual monitoring techniques.

The aforementioned issues are exemplified in the case of the invasive hornet *Vespa velutina nigrithorax*—commonly known as the Asian or Yellow-Legged hornet[10]—a eusocial vespid originating from South East Asia. This species has spread rapidly across parts of East Asia and Europe since initial colonisation events in or before 2003[11,12] and 2004[13,14], raising concern due to its consumption of native invertebrates[15,16], and predation upon colonies of the European honey bee, *Apis mellifera*[17]. Notably, *V. velutina* has also recently been recorded in North America[18], and existing invasion fronts have continued to expand despite substantial national and international management efforts[19–21]. A key reason for this is that current surveillance methods rely upon manual identification[22], and thus struggle to achieve the coverage, accuracy, and vigilance required to detect invasions

into new regions. As such, there is an urgent need to develop systems that can address these limitations, and thus bolster long-term containment strategies. Here, we demonstrate how an AI-based approach—utilising deep learning to detect and identify *V. velutina*—can provide a solution to this challenge, and hence fundamentally enhance the control of this globally invasive species.

To prevent the invasion of *V. velutina* into an area, it is essential that ingressions are detected as early as possible, enabling colony destruction before the production of new queens[10,23]. If this narrow window is missed, the population is likely to become established, and management costs will scale rapidly[24]. At present, this task depends upon visual alerts from bee-keepers and the public; however these suffer from high rates of mis-identification—yielding a mean accuracy of only 0.06% in the UK[25]. Additionally, other methods such as baited kill-traps are of limited efficacy, as they do little to reduce hornet populations, and result in considerable 'bycatch' of non-target insects[26,27]. Effective control thus requires the capture of live hornets to determine the location of the colony[10], but the initial detection of individual foragers is difficult and time-consuming. As such, an automated, accurate, and passive monitoring system is needed to improve both the sustainability and efficacy of future exclusion efforts.

The applicability of deep learning to this challenge is evident through parallel machine vision applications in behavioural tracking[28–31], pest

[1]Environment and Sustainability Institute, University of Exeter, Penryn, Cornwall TR109FE, UK. [2]Institute for Data Science and Artificial Intelligence, University of Exeter, Exeter EX44QF, UK. [3]Centre for Ecology and Conservation, University of Exeter, Penryn, Cornwall TR109FE, UK. [4]Institute of Tropical Medicine, Universitätsklinikum Tübingen, 72074 Tübingen, Germany. ✉e-mail: t.a.oshea-wheller@exeter.ac.uk

management[32–34], and conservation biology[35–37]. Consequently, there have been several attempts to develop proof-of-concept detection systems for various *Vespa* species—primarily utilising optical[38–41], infrared[42], and acoustic sensors[43]. These efforts have yielded prototype monitors for deployment at apiaries, with initial tests successfully detecting the presence of hornets in real-time[39,42]. Notably however, while such systems show promise in areas where *V. velutina* is already established, they are not well suited to provide early warning coverage along the invasion front itself. This is because initial *V. velutina* ingressions are exceedingly rare[44,45], necessitating detection models with both high precision (ensuring that only *V. velutina* are detected), and high recall (ensuring that no *V. velutina* are missed)—parameters that are often inversely related[46].

When considering the performance of current operational detection systems, these achieve mean classification accuracies within the range of ~74.5–83.3% for *V. velutina*[39,41], but suffer from false detections of other hornet species[41], and in some cases honey bees[39,42]. False positives therefore have the potential to accumulate rapidly over time, degrading the efficacy of such systems in cases where true positives are rare. Consequently, precision values consistently ≥0.99 are crucial to the successful development of a pre-emptive detection capability. The substantial complexity of this challenge, paired with the need to integrate such a capacity into compact hardware, has thus far constrained the development of a functional early alert system.

Here, we leverage deep learning to develop 'VespAI'—a remote monitor that automatically detects the presence of *V. velutina* at a specialised bait station, and relays standardised images via an automated alert. This system employs a hardware-assisted AI approach, providing state-of-the-art precision and recall performance based on YOLOv5s architecture, implemented in a compact and remotely deployable Raspberry Pi 4 processor. In addition to *V. velutina*, the algorithm can also detect and classify *Vespa crabro*—the European hornet—and robustly differentiate both species from other visually similar insects, demonstrating a precision-recall F1 score of ≥0.99. As such, VespAI has the potential to substantially improve the way in which *V. velutina* is managed, providing an accurate and passive detection capability to prevent its invasion into new regions.

## Results

### Training data pipeline
To develop an accurate but resilient detector, we employed a hardware-assisted AI approach. This utilised a standardised bait station setup, which captured training footage against a clean, uniform, and featureless background to limit environmental variation. Dependent upon local climatic conditions, it typically took up to 48 h for hornets to begin visiting these stations. Notably, the same platform was used for both training data collection and hornet detection, ensuring a consistent end-to-end pipeline. Following the collection of footage, we extracted training images that encompassed the full range of biological, environmental, and spatio-temporal variability present at bait stations, thus forming a robust dataset for annotation.

The training and validation dataset consisted of 3302 images, collected from locations across Jersey, Portugal, France, and the UK, and including a combination of three object classes: *V. velutina*, *V. crabro*, and non-target insects. The first two classes were labelled with polygonal masks to generate precise annotations for training and augmentation (Fig. 1a, b), while the third class remained unannotated, allowing the system to passively ignore all non-target species. Polygonal annotations were found to be superior to bounding boxes, as they allowed for copy-paste data augmentation[47]—in which hornets could be transposed between frames during training—subsequently improving model performance (Fig. 1c–e). Training images were specifically selected to include the range of light and weather conditions experienced by the bait stations, along with co-occurrences of both hornet species, and a diversity of non-target taxa (Table S1).

### Dataset specification
Prior to training, image data was partitioned into three subsets. The first, termed the 'hornet training subset' (HTS), consisted of 2147 images of

hornets with a limited number of non-target insects, recorded in 2021. The second, termed the 'hornet/non-target training subset' (H/NTS), consisted of 2745 images, containing a broad range of non-target insects in addition to the original hornet images from the HTS, and was also recorded in 2021. Finally, the third, referred to as the 'validation subset' (VS), consisted of 557 images, including hornets, non-target insects, and empty bait stations recorded in 2022, and was used solely for validation. Utilising this structure, we were thus able to determine how the inclusion of non-target insects influenced model robustness while ensuring that all validation data was composed of unique individual hornets.

### Model training and optimisation
Our hornet detection algorithm is built around the YOLOv5[48,49] family of instance segmentation models, with a motion and size pre-filter pipeline facilitated by ViBe[50] (Fig. 2a, S1). Fundamentally, YOLOv5 is a deep learning model based on a CSPDarkNet-53[51] CNN backbone, however it is flexible to the integration of alternate architectures[52]. We thus opted to utilise a ResNet-50[53] backbone to allow for decreased network size, although this still required a large quantity of training data to optimise the millions of potential model parameters (Fig. 2b). As such, we implemented an extensive and bespoke image augmentation routine to expand the total quantity of training data to 13,208 images, and thus expose the model to anticipated variations in image quality during inference (Fig. 1c).

Polygonal labels specifically enabled the use of copy-paste augmentation[47] (Fig. 1d), allowing us to greatly diversify the initial training data by transposing hornets between images and placing unseen individuals together. This improved both 'objectness loss' (the probability that bounding boxes contained target images) and mean average precision (the mean value of model precision across confidence thresholds). Specifically, incorporating an augmentation rate of 90% reduced objectness loss from 0.0024 to 0.0016, and increased mean average precision from 0.911 to 0.948 when compared to un-augmented models (Fig. 1e, S2a).

To enable effective functionality in the field, it was important to optimise our model for use on a remote processor. To achieve this, we compared differing base architectures to obtain a balance between performance and complexity. These consisted of the 'medium' YOLOv5m, the 'small' YOLOv5s, and the 'nano' YOLOv5n[48]. Our results demonstrated that YOLOv5m and YOLOv5s achieved comparable performance, with mean average precision values of 0.957 and 0.951 respectively, however the nano model had a reduced mean average precision of 0.934 (Fig. S2b). We thus adopted YOLOv5s as our primary architecture, enabling us to utilise the comparatively reduced model size, while maintaining performance tantamount to larger models.

### Validation and performance
Images for the validation subset were collected from different bait stations, different locations, and during a different year to the training data. This ensured that all validation images were novel, and minimised subset cross-correlation; thus facilitating a meaningful assessment of the model.

Comparisons of iterative models trained on the HTS and H/NTS demonstrated the importance of including non-target species in training (Fig. 2c, Table 1). This was characterised by differences in model F1 scores—a harmonic mean of precision and recall—as a function of acceptance threshold (Fig. 3a, Table 1). While both models exhibited high recall, their curve gradients differed at confidence thresholds within the range of 0.05-0.20—this being the region corresponding to precision. The model trained on the HTS showed F1 scores <0.95 for *V. velutina* within this region, while the H/NTS model maintained scores ≥0.99, demonstrating its improved ability to avoid false detections of non-target insects (Fig. 3a). Additionally, this latter model exhibited good performance across the recall-relevant range of confidence thresholds from 0.50 to 0.95, attaining mean F1 scores of 0.988 and 0.985 when detecting *V. velutina* and *V. crabro* respectively (Fig. 3a).

Notably, when calibrated to an optimum confidence threshold of 0.194, the final model demonstrated a true positive rate—defined as the

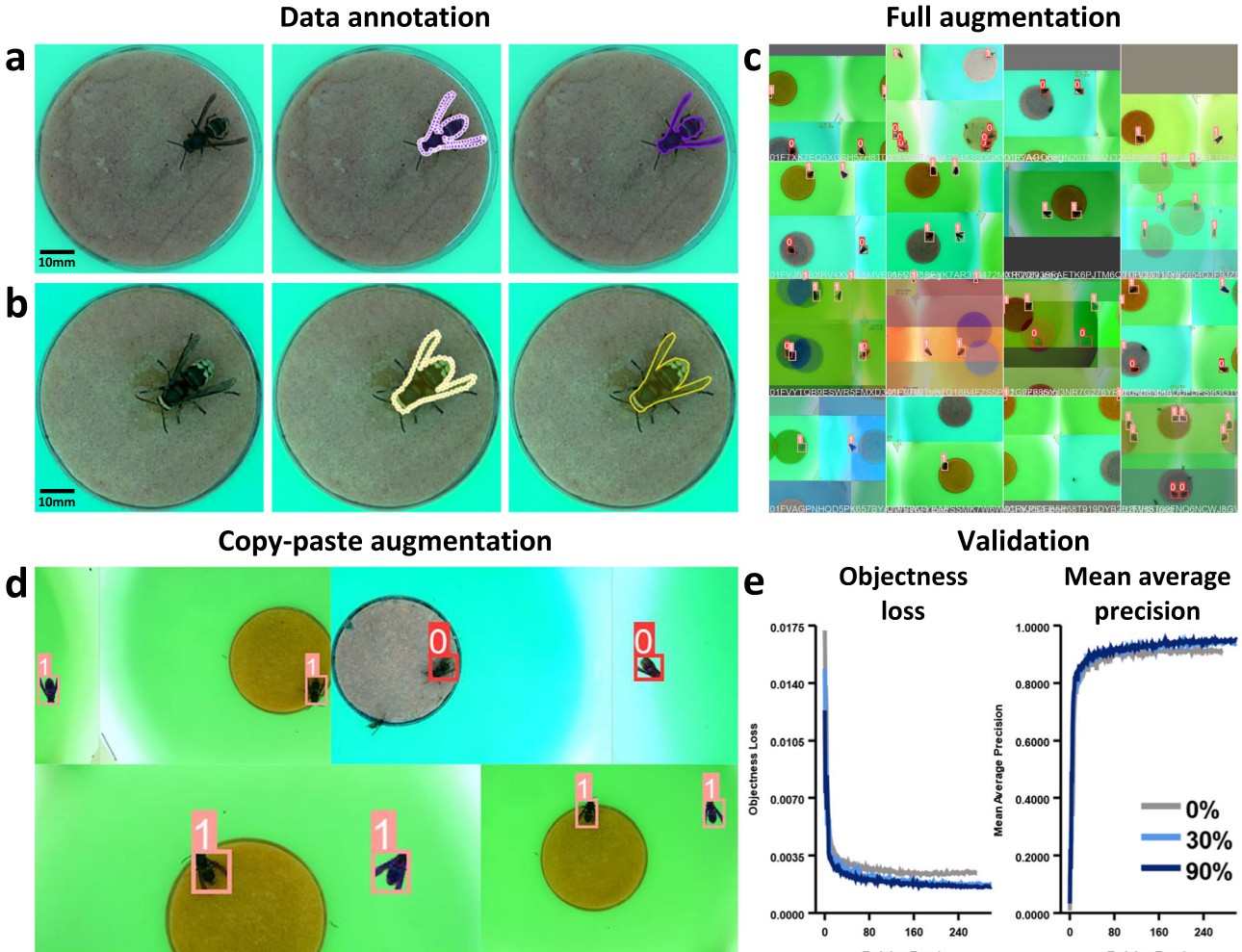

**Fig. 1 | VespAI training pipeline. a, b** Diagram of the data annotation process for **a** *V. velutina*, and **b** *V. crabro* (*V. crabro*, yellow; *V. velutina*, purple) (*N* = 3302). We employed AI-assisted annotation using Plainsight labelling software, thus producing detailed polygonal masks encompassing the body and wings of hornets. Corrections and class assignments based on species were made manually, and annotations were exported in the common objects in context (COCO)[68] format. **c** Example augmentations used to expand the training data (*N* = 13,208). These consisted of mask repositioning, image overlays, and RGB, HSV, brightness, and contrast manipulation. **d** Illustration of the copy-paste augmentation technique. Due to the use of polygonal annotations, masks could be rotated, duplicated, and moved into new frames to create novel permutations of instances. Numbers denote classes (*V. crabro*, 0; *V. velutina*, 1). **e** Performance of models with incremental increases in copy-paste training data augmentation, specifically when considering objectness loss and mean average precision for unseen validation data (*N* = 15). Line colours indicate degree of augmentation (0%, grey; 30%, light blue; 90%, dark blue).

proportion of all individual hornet detections that were correct—of ≥0.99, false positive rate of ≤0.01, true negative rate of ≥0.99, and false negative rate of ≤0.01 for both classes, translating into a combined F1 score of 0.994 (Fig. 3a, Table 1). As such, we considered this model to provide the requisite accuracy needed for effective detection of *V. velutina* incursions.

**Explaining AI predictions by pixel contribution**

Developing explainable AI models is an important consideration, as the many layers, parameters, and hyperparameters of deep neural networks pose a serious challenge to interpretation. A potential solution to this lies in the backpropagation of a model's predictions, to assess the contribution of individual pixels to a classification decision. This method is known as layer-wise relevance propagation (LRP)[54,55] and has proven to be a valuable tool both for model artefact detection, and notably, in providing insight into deep learning decision-making processes[55].

To characterise models, we thus employed an LRP-based approach to generate independent pixel-by-pixel relevance heatmaps for model classification decisions (Fig. 3b, c, and S3). We then used these predictions to identify the key visual features of both hornet species and their influence across classes and training data subsets. First, we assessed pixels contributing

to opposite and same-class predictions for the two species of hornet. Notably, we found that pixels aligned with the orange band on the fourth abdominal segment, and those around the outer edge of the wing, were important for correctly classifying *V. velutina*, but not *V. crabro* (Fig. 3b). We then investigated pixel contribution differences when achieving correct classifications with the HTS and H/NTS models, in order to determine the impact of including non-target insects. We found that the spread of relevant pixels was more focused in the H/NTS model over both classes, supporting a refinement of the deep parameters governing hornet classification (Fig. 3c).

**Prototype and deployment**

To provide proof-of-concept for remote use in the field, we produced a prototype system, integrating the software, hardware, and bait station. This setup consisted of a Raspberry Pi 4, camera, and power source, allowing the monitor to capture and analyse video in real-time (Fig. 4a), sending candidate image detections to a paired computer via a local Wi-Fi connection. We utilised this prototype design to test varying camera setups, tune hardware parameters, trial power sources, and ensure robust network connectivity (Fig. 4b and Table S2).

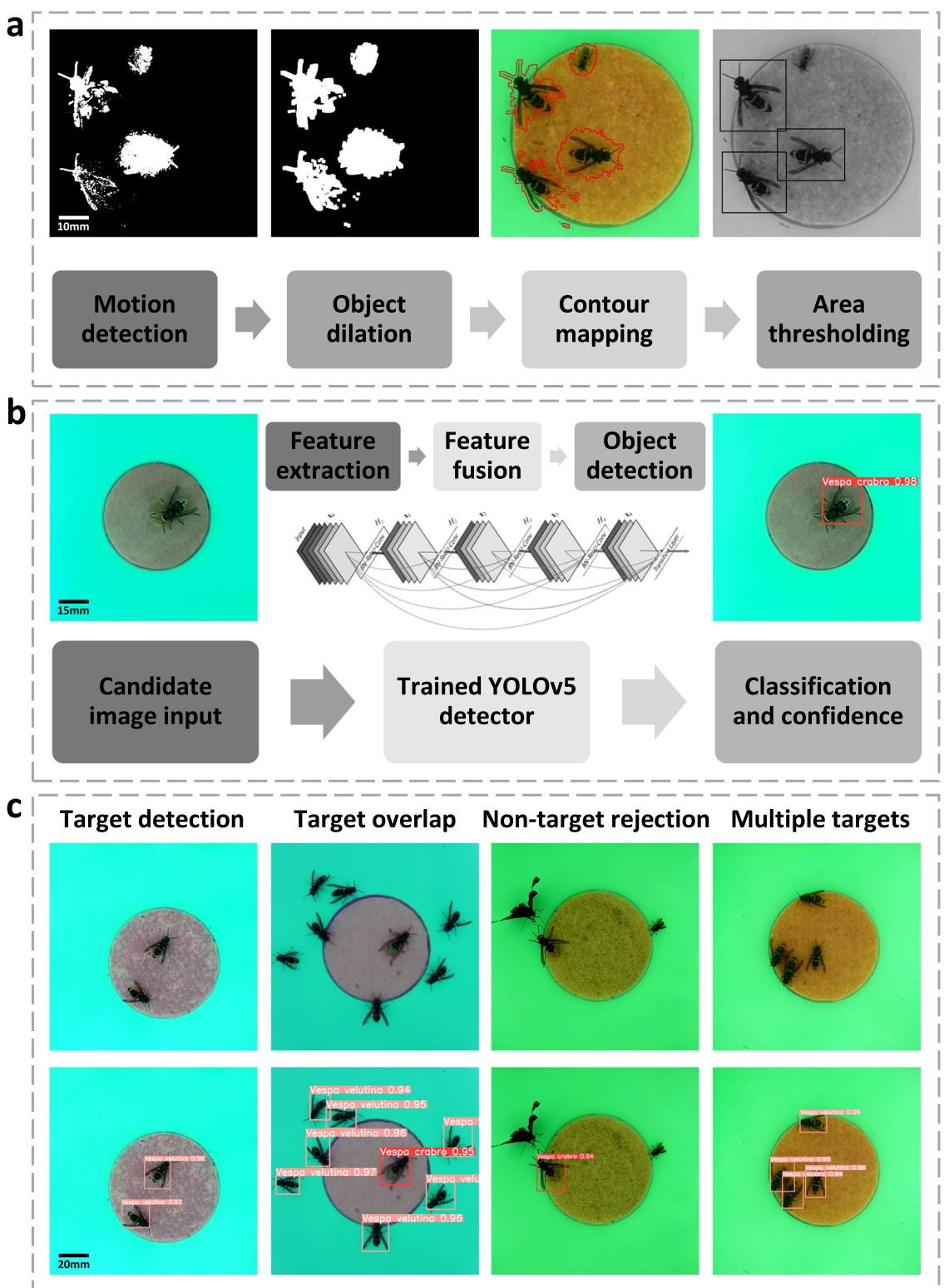

**Fig. 2 | VespAI model architecture and functionality. a** Illustration of the motion detection and video pre-filtering process used by ViBe[50]. This ensures that the system remains passive until motion is detected and that only 'hornet-sized' objects—determined from a known reference range for each species (Fig. S1)—are extracted from videos and passed on to the detection algorithm. **b** Diagram detailing the algorithm for hornet detection, classification, and confidence assignation. This model is built on YOLOv5s architecture, utilising a ResNet-50[53] backbone with a PaNet[71] neck, and applies a single F-CNN to the whole image to rapidly detect and classify hornets. To optimise performance, the algorithm downscales images to a resolution of 640 × 640 and applies letterboxing during detection. Class predictions and detection confidence values between 0 and 1 are then provided on an associated bounding box that is projected back onto the original image, as detailed in the diagram. **c** Examples of successful detections in a range of common scenarios including target saturation and overlap, class co-occurrence, and the presence of non-target insects. Dashed boxes denote discrete modules of ViBe motion detection and background subtraction, YOLOv5s object detection and classification, and example outputs when these processes are combined.

**Table 1 | Summary of evaluation metrics for models trained on the HTS and N/NTS**

| Training data | Description | Vespa velutina | | Vespa crabro | |
|---|---|---|---|---|---|
| | | mAP | F1 Score | mAP | F1 Score |
| HTS | Hornet training subset consisting primarily of hornets with few non-target insects | 0.941 | 0.989 (0.668) | 0.922 | 0.981 (0.668) |
| H/NTS | Hornet/non-target training subset consisting of hornets and a broad range of non-target insects | 0.960 | 0.996 (0.194) | 0.951 | 0.992 (0.194) |

Results are derived from model evaluation using the validation subset (VS) data, brackets indicate the confidence values at which F1 scores are calculated, mean average precision (mAP) values are calculated within the confidence range of 0.05–0.95.

VespAI functioned successfully in the field, demonstrating *V. velutina* detection performance broadly comparable to that seen during testing, with the system achieving a mean precision value ≥ 0.99 and mean recall value > 0.93 across all cameras (Fig. 4c). Notably, when considering only the highest performing camera model, these values were ≥0.99 and ≥0.96 respectively, highlighting the importance of optimally calibrated hardware (Fig. 4c). In the latter case, this translated into a true positive rate of ≥0.96, false positive rate of ≤0.01, and false negative rate of ≤0.04, as determined over 26 independent field trials (Table S3). It should be noted, however, that the aforementioned false negative rate was likely a conservative overestimate, as even when a hornet was missed in an image, the same individual was invariably detected in one of the subsequent frames.

In addition to model evaluation, field testing confirmed that hornet detections and the associated images could be sent to a paired computer or mobile device remotely, and that the system was amenable to running on a lightweight rechargeable battery for periods ~24 h. Data collected during trials also enabled the calibration of an optimised detection interval of 30 s, this being derived from minimum recorded hornet visitation durations (Fig. S4). Further work will aim to test the hardware with enhanced 4 G connectivity configurations and alternative power sources to reduce recharge frequency, while integrating the additional training data collected during field trials.

## Discussion

Our results demonstrate that VespAI achieves the requisite accuracy to detect rare *V. velutina* ingressions, while avoiding false alerts from visually similar insects. The system utilises a compact processor for deployment in the field, and a modular hardware configuration for adaptability across use cases. This fulfils the urgent operational requirement for an automated hornet detection system and advances the current state-of-the-art in invasive species monitoring. Crucially, the passive and non-lethal nature of VespAI ensures that detected hornets can be captured and tracked back to the nest for destruction, while at the same time entirely eliminating non-target bycatch. When combined with exceptionally high precision and recall—achieving an F1 score of ≥0.99—this enables pre-emptive deployment of the monitor to at-risk areas, thus helping to prevent *V. velutina* population establishment. Consequently, VespAI has the potential to transform the way in which invasive hornets are managed, providing an accurate, passive, and sustainable early detection capability to limit their spread into new regions.

The hardware-assisted AI approach utilised by VespAI enables leading-edge performance in terms of hornet detection. Specifically, the standardised bait station provides a uniform background for image capture, while training data from this same setup enables the deep ResNet architecture to achieve high identification fidelity (Fig. 3a). Upstream of this, the use of a vespid-specific attractant limits the range of visiting insects, and the ViBe filtering step ensures that only appropriately sized candidates are passed on to the YOLOv5s detection and classification algorithm. This approach differs markedly from previous systems, which aim to detect hornets against heterogenous backgrounds, and thus fail to achieve comparable levels of accuracy[38,40,56]. Indeed, the standardised solution that we employ is common to other machine learning applications in rare event detection[57–59]—a salient point when considering the scarcity of *V. velutina* colonies early in the invasion curve[24,60].

To enable effective deployment in the field, we optimised our detection model to run on a Raspberry Pi 4 processor. This hardware setup was selected to prioritise cost-effectiveness and modularity, with the ability to utilise mains and battery power sources, while providing flexible connectivity through a Wi-Fi hotspot (Fig. 4). Additionally, by opting for an edge computing solution, we aimed to negate the need for continuous connectivity to third-party infrastructure, as is the case in cloud computing approaches. Initial tests of the prototype system demonstrated the successful detection of hornets in real-time, and transfer of associated image alerts across both a local Wi-Fi network and remotely via an internet connection. Crucially, additional field testing confirmed the ability of VespAI to achieve comparable accuracy to that seen during initial model validation, and thus function effectively in regions where both *V. velutina* and *V. crabro* are present (Fig. 4c, d, and Table S3). This is pertinent when considering future operational deployment, as it demonstrates the tractability of our integrated hardware setup in the field. Depending on the specific use case, the platform can further be configured to integrate solar, battery, or mains power; and send detection alerts via direct access, Wi-Fi, or through SMS. Such versatility is an important feature when considering the breadth of potential operators, from beekeepers that often prioritise accessibility at a local scale, to governments and agencies that may wish to utilise large-scale detection networks.

While the principle aim of the VespAI system is to enhance *V. velutina* detection capabilities, it also promises to substantially reduce the environmental impact of such surveillance. This is most pertinent when considering that the majority of current monitoring strategies rely upon baited traps, which primarily capture and kill non-target insects, making them antithetical to the aims of ecosystem protection[26,27]. Further, trapping has limited efficacy in the context of control when not directed at emerging queens, as for social vespids, killing even considerable numbers of workers has little impact on the success of the colony[61,62]. VespAI is thus optimised to facilitate the detection and tracking of live hornet workers, which currently constitutes the most effective method of nest location and destruction, and forms the basis of all efficacious exclusion strategies[10]. Beyond *V. velutina*, the platform also provides a tool for the detection of *V. crabro* in areas where it is itself invasive[63] or protected[64], and its efficient training pipeline is amenable to the inclusion of additional invasive or conserved species. As such, the system has strong potential for rapid adaptation to emerging biosecurity and conservation challenges, with future development aiming to support this goal.

In conclusion, we demonstrate that VespAI provides an efficient, sustainable, and cost-effective system for the automated detection of invasive hornets. This fulfils an urgent and timely requirement in the control of *V. velutina*, with the potential to substantially improve exclusion efforts across the invasion front. Future work will aim to enhance the capabilities of VespAI through the development of an improved user interface, integration of mobile network connectivity, and expansion of the detection algorithm to encompass additional species such as *V. orientalis* and *V. mandarinia*. Taken together, our results demonstrate the power of machine vision when applied to species monitoring, and provide a robust tool for future control, conservation, and research applications.

## Methods
### Bait station
Bait stations consisted of a Dragon Touch Vision 1 1080p camera, suspended at a height of 210 mm above a featureless detection board, shielded by an opaque baffle (Fig. 4). This setup minimised background and lighting

**Fig. 3 | VespAI model optimisation and explanation. a** F1 confidence curves demonstrating mean precision and recall for models trained on the hornet training subset (HTS), and hornet/non-target training subset (H/NTS), tested against the unseen validation subset from 2022. Line colours indicate classes (*V. crabro*, yellow; *V. velutina*, purple; class mean, blue), with the left side of the confidence axis predominately corresponding to model precision, and the right side to recall. **b** Layer-wise relevance propagation heatmaps demonstrating pixel relevance when classifying hornets, divided by class, with dark areas indicating regions of irrelevance to the class in question. The opposite class panels highlight relevant pixels leading to the wrong class prediction, while the same-class panels highlight relevant pixels leading to a correct decision. The brightness of the pixels is normalised across images to demonstrate the comparative relevance distributions of specific image regions. **c** Layer-wise relevance propagation heatmaps demonstrating pixel relevance when classifying hornets, divided by model training data. The HTS panels highlight relevant pixels when classifying hornets with the hornet training subset model, while the H/NTS panels detail this for the hornet/non-target training subset model.

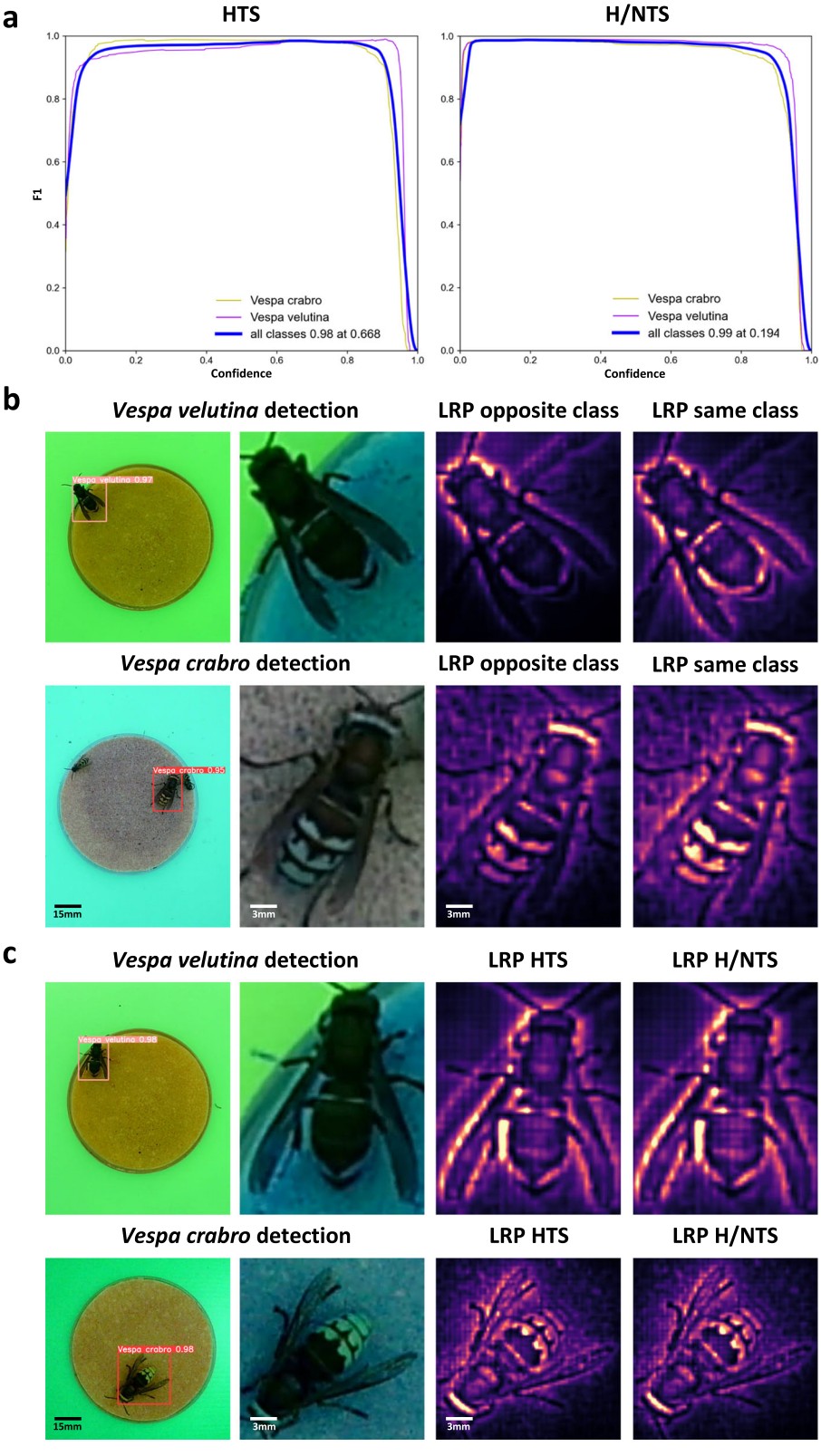

variability, thus simplifying the computational complexity of hornet detection, while ensuring that only hornets and other insects visiting the station were captured in videos. A sponge cloth impregnated with commercial vespid attractant—VespaCatch (Véto-pharma) or Trappit (Agrisense)—was placed in a 90 mm diameter Petri dish at the centre of the bait station, thus attracting hornets to land directly beneath the camera. We used these bait stations to collect and extract an extensive training dataset, comprising images of *V. velutina*, *V. crabro*, and other insects across locations in Jersey, Portugal, France, and the UK.

To ensure dataset fidelity, resultant images of both *V. velutina* and *V. crabro* were visually identified via expert assessment of colouration, abdominal markings, and morphology. Additionally, the identity of each hornet species was confirmed through utilisation of the appropriate taxonomic keys[65,66].

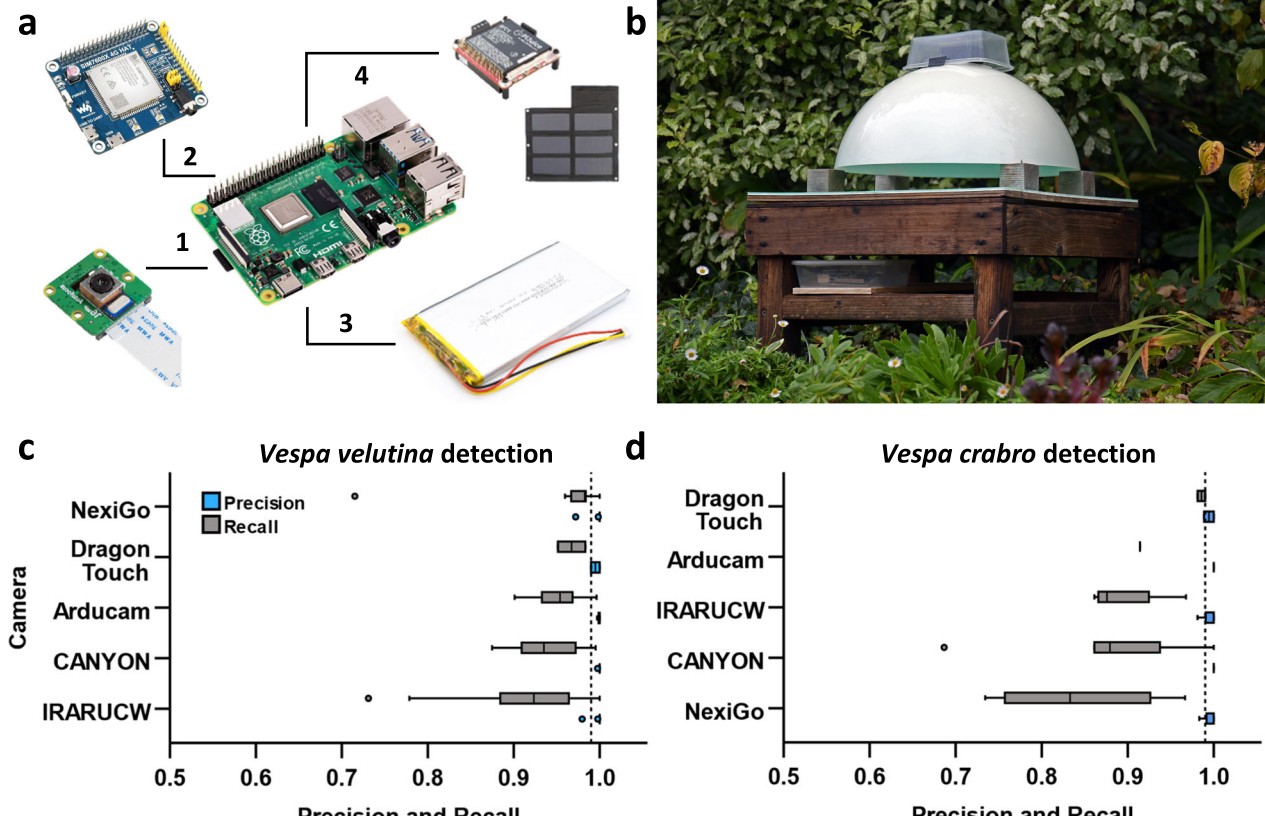

**Fig. 4 | VespAI hardware and performance. a** Diagram of components and optional additions for the detector hardware. The system is built around a Raspberry Pi 4, with flexible modular components including (1) a 16MP IMX519 autofocus camera module; (2) a 4 G HAT with GNSS positioning for remote transmission of detections via SMS; (3) a PiJuice 12,000 mAh Battery; and (4) a PiJuice 40 watt solar panel for self-sustaining remote deployment. The hardware configuration is not limited to these components and will work with any Raspberry Pi 4-compatible additions, allowing for complete customisation based on use case and budget. Photographs courtesy of Raspberry Pi Ltd. **b** Prototype setup of bait station and hardware to test the VespAI algorithm in the field. **c, d** Precision and recall scores across candidate cameras for **c** *V. velutina* and **d** *V. crabro* during field testing of the prototype system (*N* = 55). Boxplots are coloured by measure (precision, blue; recall, grey) and grouped by camera performance. Dashed lines indicate the desirable precision threshold of >0.99. Outliers (greater than 1.5 times the interquartile range from the median) are denoted with circles.

## Data collection

Data were collected in 2021 and 2022, with selected images being extracted from the raw video footage, and divided into three subsets. All training images were collected in 2021, while the final validation images were collected in 2022, ensuring complete spatiotemporal and biological novelty. Images yielded a maximum simultaneous co-occurrence of six *V. velutina*, this being observed in Jersey; and five *V. crabro*, this being recorded in the UK. As a processing step prior to training, images were letterboxed—this being the process of downsampling to 640 × 640 for enhanced throughput performance, while maintaining a 16:9 aspect ratio and filling any residual image space with blank pixels. This then allowed for extensive image augmentation during training, producing three additional variations to supplement each original frame, and thus increasing the total number of images by a factor of four. The specific details of each training data subset are outlined in the following sections.

**Hornet training subset (HTS).** A collection of 1717 images for training and 430 for initial validation metrics, totalling 8,588 after augmentation. This set contained hornet images with a 50:50 split between *V. velutina* and *V. crabro*, while the number of non-target insects was intentionally limited. Data were collected from bait stations at sites in the UK and Portugal.

**Hornet/non-target training subset (H/NTS).** A collection of 2196 images for training and 549 for initial validation metrics, totalling 10,980 after augmentation. This set contained all hornet images from the HTS, in

addition to 598 images of non-target insects. Images of non-target insects included a representative selection of species attracted to the bait station, with a focus on visually similar genera such as *Vespula*, *Dolichovespula*, and *Polistes*. All insects were identified to the genus level, utilising a combination of expert assessment and the relevant taxonomic identification resources[65,67]. A full list of non-target taxa is provided in (Table S1). These data were collected from bait stations at sites in the UK, Jersey, and Portugal.

**Validation subset (VS).** A collection of 557 images for final validation only, totalling 2228 after augmentation. Of these, 433 contained instances of *V. velutina* and *V. crabro* in a 50:50 split, including multiple co-occurrences of both species and non-target insects. The remaining images contained a combination of non-target species and empty bait stations under different lighting and climatic conditions. Validation data were collected from bait stations at sites in the UK, Jersey, France, and Portugal.

## Data annotation

Annotation was performed using the Plainsight AI (Plainsight) software interface. This allowed for expedited labelling via automated polygon selection and AI-assisted predictive annotation. Two classes of annotation were generated, corresponding to *V. velutina* and *V. crabro*, and these were then manually applied to a random selection of training frames. Polygonal masks included hornet bodies and wings, and excluded legs and antennae—as we found these to be redundant during testing. Once ~500 frames had

been annotated manually, we then used this data to train an automated detection and segmentation model within the labelling interface, allowing us to more rapidly generate further annotations for training. Prior to data export, all annotations were reviewed manually, and corrections made where required. Annotations were exported in COCO format, enabling full segmentation of hornet features from the background[68].

## VespAI software

To develop a hardware-specific hornet detection and classification model, we combined our extensive image dataset with bespoke augmentations to obtain high predictive confidence. The VespAI detection algorithm is built on the YOLOv5 family of machine vision models, specifically YOLOv5s—a variant optimised to run on portable processors such as the Raspberry Pi 4[48]. As a front-end pre-filter to this, we incorporated the lightweight ViBe[50] background subtraction algorithm, allowing the system to remain passive in the absence of motion (Fig. 2a). Specifically, this pre-filter detects motion from the raw video input, extracts the contours of moving insects, and retains only objects within a reference size range generated from known hornet detections (Fig. 2a and S1). Consequently, energy is conserved, as only relevant candidate frames are passed on to the YOLOv5 detection algorithm itself. This then applies a single fully convolutional neural network (F-CNN) to images (Fig. 2b), providing superior speed, accuracy, and contextual awareness when compared to traditional regional convolutional neural networks (R-CNN)[49,69].

All models were built and optimised using the PyTorch[70] machine learning environment, with the aim of generating an end-to-end software package that would run on a Raspberry Pi 4. This was achieved by testing models on a range of YOLOv5 architectures, specifically YOLOv5m, YOLOv5s, and YOLOv5n; thus optimising them to include the minimum number of parameters—this being ~7 million—whilst maintaining their performance (Fig. S2b).

Final models were trained and tested utilising a NVIDIA Tesla V100 Tensor Core GPU (NVIDIA), with a total of 200–300 epochs per model, and a batch size of nine images. Model optimisation was evaluated via three loss functions; bounding box loss, this being the difference between the predicted and manually annotated bounding boxes; objectness loss, defined as the probability that bounding boxes contained target images; and cross-entropy classification loss, encompassing the probability that image classes were correctly classified (Fig. S2). In all cases, training concluded when there was no improvement in these three loss functions for a period of 50 epochs.

## Prototype hardware

The prototype system was developed to provide proof-of-concept for remote detection under field-realistic conditions. The VespAI software was installed on a Raspberry Pi 4, running an Ubuntu desktop 22.04.1 LTS 64-bit operating system. This was then connected via USB to a variety of 1080p cameras, and tested using both mains and battery power supplies. These components were mounted on top of a bait station in the standard camera position, and a remote device was connected to the Pi server via the secure shell command. This allowed the hardware to be controlled remotely, and hornet detections viewed from a corresponding computer.

The setup was validated in Jersey during 2023, testing five candidate camera models and four prototype systems over a total of 55 trials at two field sites, yielding >5500 frames for analysis. Cameras were selected to test system robustness to differing lens and sensor options, while maintaining a standard resolution of 1080p across a range of cost-effective models (Fig. S5 and Table S2). Prior to testing, each camera was calibrated to a specific height, thus ensuring that the relative size of objects in frame remained constant across differences in lens angle and focal length (Table S2). Field sites were situated in Jersey to allow visits from both *V. velutina*, and *V. crabro* workers, along with a variety of common non-target insects, thus providing a rigorous test of the system under representative conditions.

Each trial consisted of a ≥ 100-frame test, with the monitor capturing and analysing frames in real-time at intervals of either 5 or 30 s—these being based on known hornet visitation durations (Fig. S4). Specifically, in the first

38 trials, the system was set to collect images at 5 s intervals; before optimising to 30 s intervals in the final 17 trials (Table S3), thus allowing for maximum power and data storage conservation, in tandem with reliable hornet detection. (Fig. S4). Results were then manually validated, and compared to the corresponding model predictions to calculate evaluation metrics.

Following field testing, the system was configured to integrate a DS3231 Real-Time Clock module, thus ensuring accurate timestamps for detections in the absence of external calibration.

## Statistical analyses

To train the detection models and enable customised image augmentation, we employed the Python packages 'PyTorch', 'Torchvision', and 'Albumentations'. Models were then evaluated via k-fold cross-validation, specifically utilising the metrics of precision, recall, box loss, objectness loss, classification loss, mean average precision (mAP), and F1 score (Fig. S2 and Table 1). Cross-validation analyses employed a subsample (k) of 5, as this proved sufficient to select an optimised detection classifier that balanced model size with performance. Resultant model rankings were based on mean cross-validation scores, calculated using the Python packages 'scikit-learn' and 'PaddlePaddle', and the 'YOLOv5' integrated validation functionality. Additional performance visualisations were generated via the packages 'Seaborn', 'Matplotlib', and 'NumPy'. All statistical analyses were performed in SPSS (release v. 28.0.1.1) and Python (release v. 3.9.12).

**Training data pipeline.** Cross-validation of polygonal and box annotation techniques utilised precision, recall, box loss, objectness loss, classification loss, and mAP as response variables, and compared models with copy-paste augmentation levels of 0%, 30%, and 90%, with the former of these corresponding to box annotations.

**Dataset specification.** Visualisation of training data subsets to ensure sufficient image novelty utilised frequency distribution analyses of blur, area, brightness, colour, and object density between the HTS, H/NTS, and VS.

**Model training and optimisation.** Cross-validation of model architectures employed precision, recall, box loss, objectness loss, classification loss, and mAP as response variables, and compared models using the YOLOv5m, YOLOv5s, and YOLOv5n architectures.

**Validation and performance.** Cross-validation of models trained on the hornet training subset and hornet/non-target training subset used F1 score and mAP as response variables, and compared models trained on the HTS and H/NTS, validated against the VS.

**Explaining AI predictions by pixel contribution.** The LRP class classification model employed normalised contributions to classification decisions as a response variable, and compared same and opposite class pixel contributions. The LRP training subset classification model used normalised contributions to classification decisions as a response variable, and compared models trained on the HTS and N/HTS.

**Prototype and deployment.** Precision and recall analyses were utilised to compare camera models, with comparisons based on median performance across test types for each metric.

## Statistics and reproducibility

Model development utilised a sample of 3302 images collected from a total of four countries, each consisting of multiple sampling sites. Data augmentation further expanded this sample to 13,208 images and provided additional variation to enhance model robustness. Analyses of the prototype system employed a sample of >5500 frames, collected across 55 field trials at two sites in Jersey. The source data underlying all figures and analyses are available within the supplementary data. Full details of statistical tests, subset sample sizes, and model selection procedures are provided in the results and statistical analyses sections.

**Reporting summary**

Further information on research design is available in the Nature Portfolio Reporting Summary linked to this article.

## Data availability

The authors declare that all supporting data is available within the supplementary information. For source data underlying the field trial figures and analyses, see (Supplementary Data).

## Code availability

All model code, validation data, manuals, and hardware setup instructions are available under a CC BY-NC-SA 4.0 license at: https://github.com/andrw3000/vespai. This permits usage and adaptation for non-commercial applications, with any derivatives falling under the same restrictions. Access to this data must be requested via contacting the corresponding author and providing a statement outlining its intended use case. This pathway aims to prevent unauthorised commercial usage, while facilitating research collaboration. All such requests will receive a response within 14 days.

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

## Acknowledgements

T.A.O.-W, J.L.O, P.J.K, and A.C were funded by the UKRI Biotechnology and Biological Sciences Research Council (BBSRC) (Project No. BB/S015523/1), with further support from a University of Exeter IDSAI Research Award. We would like to thank Alastair Christie (Asian hornet co-ordinator, States of Jersey), members of the Jersey Asian Hornet Group, Marco Portocarrero (Associação NATIVA), and Miguel Maia (Apis Maia) for helping with data collection. We also thank the Raspberry Pi Foundation for providing images of hardware components, and Andre Koch, Jim Williams (University of Exeter, Innovation, Impact & Business team), and Prof Andy King (University of Kent) for providing useful discussions.

## Author contributions

T.A.O.-W., J.L.O., M.R., and P.J.K. conceived the study; T.A.O.-W. wrote the manuscript; T.A.O.-W. and P.J.K. collected the data; A.C. developed and integrated the model; T.A.O.-W. and A.C. carried out the statistical analyses. All authors edited the manuscript and gave final approval for publication.

## Competing interests

The authors declare no competing interests.
