## [Peer Review File · Communications Biology]

Reviewers' comments:

Reviewer #1 (Remarks to the Author):

The article proposes an automated system for the rapid detection of *V. velutina* based on deep learning methods. The paper is well written and detailed. However, some important details and information needs to be added.

Abstract: brief and explanatory. Can provide more detail on the methods used, the main conclusions. May also refer to possible future work.

Introduction: Context very well explained. Well referenced. A review of previous works is lacking, with a more technical framing of the AI methods used in this field.

Lines 58 – 60: Why can't they provide in the operational monitor? Substantiate this sentence.

Lines 60 – 64: Add a justification why they don't work for detecting *Vespa* species. There are several scientific articles that prove that it is possible to detect insects of very similar classes and of very small dimensions with a high degree of confidence. Why doesn't it work for detecting *Vespa* or is there any difficulty?

In this section there is a gap regarding the state of the art. Metrics and analysis of other work carried out in this context must be provided. They may also include a paragraph justifying their choice to use YOLOv5. May also mention the use of applied AI explaining.

Results:

Line 90 : State that the polygonal annotation allowed for a wider array of data augmentation, and improved model performance, however, they do not prove this assertion nor do they refer to works in which this was observed.

Explanation of methodology very confusing. Needs improvements in explaining the methodologies used. You used a classification model with F-CNN and the YOLOv5 detector, a bit confusing.

Line 112 – 115: Was the applied classification model an F-CNN?

Line: 116: YOLOv5 is an object detection architecture that consists of Backbone, Neck and Head. On the backbone, default is CSPDarknet-53. Therefore, it is not understood where can claim that "YOLOv5 is a deep learning model based on a ResNet 50 CNN backbone". If used ResNet50 instead of CSPDarknet-53, should say so more clearly.

How many images does the dataset have after data augmentation?

In the results, they can provide the values obtained by the metrics such as recall, precision, mean average precision, recall – precision curve and add a table with the summary of the results.

Materials and Methods:

Line 278: classifier or detector (locate and classify)

Make a detailed description of the libraries and packages used but do not mention important training features, such as the number of epochs, batch size, loss function. So, the used loss function and important training parameters should be mentioned.

Reviewer #2 (Remarks to the Author):

The article describes an automated, AI-based detection system for invasive hornets. The article is well written, original and as far as I can judge, without scientific flaws (but see my comments to the statistical analysis section). The detection system is furthermore much needed for the management of the yellow-legged Asian hornet *Vespa velutina nigrithorax* and an excellent step forward for early detection in future invasions. While I cannot judge the use of AI (which nevertheless seems very sound to me), I can make some suggestions for clarifications from the point of view of an entomologist and invasion ecologist. Therefore, I ask the authors to consider the following points:

- Lines 20 and 71: "precision-recall confidence of ≥ 0.99 " Is there a unit for this value?

- Line 30: I am not sure what is meant by "repeated but rare founding events", could this be described in a bit more detail for a better understanding?
- Line 35: This may be a detail, but the common understanding seems to be that the yellow-legged hornet was detected first in 2004, not 2005. But the cited authors seem to be contradicting themselves in various articles. It would be good to confirm the first year of detection (e.g. from the literature or by consulting French specialists).
- Lines 74 ff.: Every sub-section of the results starts with a rather detailed description of the material and methods, which is then repeated in the Materials and Methods section. I do not know if this is a requirement of the journal, but it seems to be unnecessary. Please consider to at least shorten these sections.
- Line 121: I am not familiar with the word "letterboxed". Please consider to briefly describe what is exactly meant by that.
- Line 126: In my opinion the term "objectness lost" should be explained.
- Line 137: It is unclear what is meant by "new locations", I can assume that it is different locations than for the training, but this should be more clearly stated please.
- Line 210: The authors talk about "a variety of power sources and network options" but only one seems to be presented, please specify or correct.
- Line 238 ff: What is clearly missing from the Materials and Methods is the description of how the insects were morphologically identified before presented to the AI. This may be straightforward for an entomologist but I think that the manuscript is aimed for a broader audience and it should be explained how a thorough identification of the insects can be done (e.g. by citing identification keys).
- Lines 256 and 263: How were these species and genera identified? Taxonomic keys and knowledge are needed which is not described here.
- Lines 296 ff: It is unclear which statistical tests were conducted. The authors simply state that the metrics were assessed but the tests employed for the assessment are not described. Please describe in full detail.
- Lines 299 f.: The following terms have to be explained somewhere, otherwise it is not possible for the reader to understand the statistics that were done: "precision, recall, box loss, objectness loss, classification loss, mean average precision (mAP), and F1 confidence scores".
- Lines 345 ff.: The caption of figure 3 is unclear. Abbreviations like F1, VS, and LRP are used that cannot be understood in the context. Also, what is the difference between pixel relevance and pixel importance, please clarify or make it uniform.

Response to Referees

Editing Key:

Corresponding lines in revised manuscript-**yellow**, Replies-**blue**

Editorial Board Member, Dr Luke R. Grinham

Comments to Author:

Your manuscript entitled "VespAI: A Deep Learning-Based System for the Detection of Invasive Hornets" has now been seen by 2 referees. You will see from their comments below that while they find your work of considerable interest, some important points are raised. We are interested in the possibility of publishing your study in *Communications Biology*, but would like to consider your response to these concerns in the form of a revised manuscript before we make a final decision on publication.

Reviewer(s)' Comments to Author:

Referee: 1

The article proposes an automated system for the rapid detection of *V. velutina* based on deep learning methods. The paper is well written and detailed. However, some important details and information needs to be added.

Thank you for these constructive comments.

Abstract: brief and explanatory. Can provide more detail on the methods used, the main conclusions. May also refer to possible future work.

We have now expanded the abstract to better encompass the use of YOLOv5s, along with briefly referencing potential future work. See lines: **14-26.**

Introduction: Context very well explained. Well referenced. A review of previous works is lacking, with a more technical framing of the AI methods used in this field.

We have added a more comprehensive review of previous work, specifically with a focus on the AI methodologies and hardware setups employed. See lines: **61-64.**

Lines 58 – 60: Why can't they provide in the operational monitor? Substantiate this sentence.

We have now clarified this statement to refer specifically to an operational early warning system, and have explicitly outlined the additional challenges that this poses for current monitors. Detection of *V. velutina* during initial incursions requires a system with very high precision to avoid the accumulation of false positives, as true positives are likely to be rare. The majority of studies provide proof-of-concept through training and validation image sets, and there are two studies that demonstrate prototype systems, however these are designed for use in the lab, and deployment at apiaries in invaded areas. As such, they are not equipped to deliver the detection precision and recall needed for use along the invasion front itself. See lines: **61-79.**

Lines 60 – 64: Add a justification why they don't work for detecting *Vespa* species. There are several scientific articles that prove that it is possible to detect insects of very similar classes and of very small dimensions with a high degree of confidence. Why doesn't it work for detecting *Vespa* or is there any difficulty?

While previous work has shown that *V. velutina* can be detected successfully via machine-learning, current working systems achieve accuracy values in the range of 74.5-83.3%, which are generally insufficient to avoid multiple false positives. This is pertinent when running a system continuously in areas where *V. velutina* are rare—but similar insects are common—such as in those regions along the invasion front, or where ingressions have not yet occurred. The challenge of an early alert system is thus one that requires accurate rare event detection, but with few to no false positives, as these have the potential to accumulate rapidly. For example, in (Herrera et al. 2023), 26.1% of *Vespa crabro* and 11.4% of *Polistes dominula* were erroneously classified as *V. velutina*, which would substantially degrade the system's ability to function as an effective alert platform. We have now provided a more comprehensive explanation of this difficulty, and referenced specific examples and performance metrics. See lines: 61-79.

In this section there is a gap regarding the state of the art. Metrics and analysis of other work carried out in this context must be provided. They may also include a paragraph justifying their choice to use YOLOv5. May also mention the use of applied AI explaining.

We have now included further discussion of the current state-of-the-art, including metrics and evaluation of existing AI-based hornet detection systems. See lines: 72-76. While it is beyond the scope of our introduction, we have provided an extensive justification of our decision to use YOLOv5 in the results and methods sections. See lines: 120-127, 315-326.

Results:

Line 90 : State that the polygonal annotation allowed for a wider array of data augmentation, and improved model performance, however, they do not prove this assertion nor do they refer to works in which this was observed.

We have clarified this statement to detail that polygonal masks allowed for copy-paste augmentation during training, in which hornets can be moved between images to generate novel combinations. This assertion is supported by Fig. 1e, that details improved model performance with increasing increments of copy-paste augmentation during training. See figure: 1. Additionally, we have provided a reference outlining the copy-paste augmentation procedure, and its ability to enhance model training. See lines: 103-106, 386-389.

Explanation of methodology very confusing. Needs improvements in explaining the methodologies used. You used a classification model with F-CNN and the YOLOv5 detector, a bit confusing.

All YOLO models employ fully convolutional neural networks (F-CNNs) for classification and image segmentation, which differentiates them from the majority of other models in computer vision, and provides improved speed and accuracy. Specifically, this means that YOLO analyses the entire image once, as a single regression problem, rather than performing multiple predictions on different image regions. We have now added detail to crystallise this in the methodology, and reduce confusion. See lines: 324-326.

Line 112 – 115: Was the applied classification model an F-CNN?

Yes, we have reworded to make this clear. See lines: 324-326.

Line: 116: YOLOv5 is an object detection architecture that consists of Backbone, Neck and Head. On the backbone, default is CSPDarknet-53. Therefore, it is not understood where can claim that "YOLOv5 is a deep learning model based on a ResNet 50 CNN backbone". If used ResNet50 instead of CSPDarknet-53, should say so more clearly.

This is a salient point, we have subsequently clarified the statement to specify that we selected a ResNet-50 backbone instead of the default CSPDarknet-53 backbone due to its smaller network size. See lines: 121-124.

How many images does the dataset have after data augmentation?

Augmentations yielded three variations per image in addition to the original, for a total of 13,208 images in the augmented dataset. We have now included this information, along with specific image numbers for each dataset. See lines: 126-127, 285-286, 290-291, 298.

In the results, they can provide the values obtained by the metrics such as recall, precision, mean average precision, recall – precision curve and add a table with the summary of the results.

We have added a table to the results summarising these metrics. See table: 1.

Materials and Methods:

Line 278: classifier or detector (locate and classify)

This is a relevant point, the model detected and then segmented hornets in images to generate additional polygons. We have now corrected this, along with checking the use of such terminology throughout. See lines: 309-310, 315, 324, 340, 394-395.

Make a detailed description of the libraries and packages used but do not mention important training features, such as the number of epochs, batch size, loss function. So, the used loss function and important training parameters should be mentioned.

We have now provided details of the key training features as suggested, along with the hardware used for training, and the specific loss functions employed. See lines: 332-338.

Referee: 2

The article describes an automated, AI-based detection system for invasive hornets. The article is well written, original and as far as I can judge, without scientific flaws (but see my comments to the statistical analysis section). The detection system is furthermore much needed for the management of the yellow-legged Asian hornet *Vespa velutina nigrithorax* and an excellent step forward for early detection in future invasions. While I cannot judge the use of AI (which nevertheless seems very sound to me), I can make some suggestions for clarifications from the point of view of an entomologist and invasion ecologist. Therefore, I ask the authors to consider the following points:

Thank you for these positive comments.

- Lines 20 and 71: “precision-recall confidence of ≥ 0.99 ” Is there a unit for this value?

The unit for this value is the F1 score, which is a mean of precision and recall. We have subsequently specified this as suggested. See lines: 21-22, 86.

- Line 30: I am not sure what is meant by “repeated but rare founding events”, could this be described in a bit more detail for a better understanding?

We have now clarified this in the context of queen dispersal. Specifically, queens can repeatedly disperse into new regions adjacent to invaded areas, however because such events may comprise relatively few reproductive individuals, their overall prevalence is comparatively low, making detection difficult. See lines: 33-34.

- Line 35: This may be a detail, but the common understanding seems to be that the yellow-legged hornet was detected first in 2004, not 2005. But the cited authors seem to be contradicting themselves in various articles. It would be good to confirm the first year of detection (e.g. from the literature or by consulting French specialists).

Thank you for highlighting the apparent contradiction between citations. We have modified the references to clarify that *V. velutina* was first officially detected and identified in 2005, but following an extensive survey in 2006, anecdotal reports were received from French citizens that suggested *V. velutina* was likely present in the Lot-et-Garonne area in 2004. Consequently, French researchers refer to *V. velutina* being first detected in 2005, but likely arriving in or before 2004. We have now corrected this to refer to the dates of first introduction to avoid confusion, and updated the corresponding references. See line: 39.

- Lines 74 ff.: Every sub-section of the results starts with a rather detailed description of the material and methods, which is then repeated in the Materials and Methods section. I do not know if this is a requirement of the journal, but it seems to be unnecessary. Please consider to at least shorten these sections.

While we understand this point, it is the requirement of *Communications Biology* to include sufficient methodological details in the results section, along with expanded information in the methods themselves. We have, however, attempted to make the results more concise by moving some of the current information into the methods section. See lines: 99-108, 120-134, 181-193.

- Line 121: I am not familiar with the word "letterboxed". Please consider to briefly describe what is exactly meant by that.

Letterboxing is the process of resizing images to different resolutions while maintaining their original aspect ratios by filling any empty space with blank pixels. We have now clarified this. See lines: 278-281.

- Line 126: In my opinion the term "objectness lost" should be explained.

This is a reasonable point, we have now provided a more detailed definition. See lines: 130-131, 335-336.

- Line 137: It is unclear what is meant by "new locations", I can assume that it is different locations than for the training, but this should be more clearly stated please.

That is indeed what is meant, we have subsequently reworded the sentence to make this clear. See line: 144.

- Line 210: The authors talk about "a variety of power sources and network options" but only one seems to be presented, please specify or correct.

We have corrected this to specify the use of both mains and battery power, along with a local Wi-Fi hotspot and internet connection. This section has also now been enhanced to provide full details of the field trials in which power sources and network options were tested. See lines: 224-237.

- Line 238 ff: What is clearly missing from the Materials and Methods is the description of how the insects were morphologically identified before presented to the AI. This may be straightforward for an entomologist but I think that the manuscript is aimed for a broader audience and it should be explained how a thorough identification of the insects can be done (e.g. by citing identification keys).

That is a salient point, we have now detailed this and provided references to the appropriate taxonomic identification keys. See lines: 271-272.

- Lines 256 and 263: How where these species and genera identified? Taxonomic keys and knowledge are needed which is not described here.

We have subsequently provided details outlining this, along with references to the relevant identification resources. See lines: 294-295.

- Lines 296 ff: It is unclear which statistical tests were conducted. The authors simply state that the metrics were assessed but the tests employed for the assessment are not described. Please describe in full detail.

We employed k-fold cross-validation to evaluate and select models. We have now specified this, and provided details of the subset size and ranking procedure. See lines: 341-347.

- Lines 299 f.: The following terms have to be explained somewhere, otherwise it is not possible for the reader to understand the statistics that were done: “precision, recall, box loss, objectness loss, classification loss, mean average recision (mAP), and F1 confidence scores”.

We have now defined each term upon first use. See lines: 69, 70, 130-132, 149, 334-337.

- Lines 345 ff.: The caption of figure 3 is unclear. Abbreviations like F1, VS, and LRP are used that cannot be understood in the context. Also, what is the difference between pixel relevance and pixel importance, please clarify or make it uniform.

We have addressed this as suggested, defining each term in full within the caption, and standardising the terminology to refer to pixel relevance rather than importance in all cases. See lines: 404-416.

REVIEWERS' COMMENTS:

Reviewer #1 (Remarks to the Author):

The authors responded and followed all recommendations given.

Reviewer #2 (Remarks to the Author):

Dear Authors,

Dear Editor,

All of my suggestions were incorporated into the article with great care and additional explanation in the answers to the reviewers. I therefore do not see the necessity for further changes and can recommend the publication of the manuscript.

Congratulations to the authors for this very interesting and useful work!